# Phenotype stability and dynamics of transposable elements in a strain of the microalga *Tisochrysis lutea* with improved lipid traits

**Jérémy Berthelier**[1,2,3]*, **Bruno Saint-Jean**[1,4], **Nathalie Casse**[2], **Gaël Bougaran**[1,5], **Grégory Carrier**[1,4]

**1** Laboratory PBA, Ifremer, Nantes, France, **2** Laboratory of Biology of Organisms Stress Health Environment (BiOSSE), Le Mans University, Le Mans, France, **3** Plant Epigenetics Unit, Okinawa Institute of Science and Technology Graduate University, Onna, Okinawa, Japan, **4** Laboratory PHYTOX GENALG, Ifremer, Nantes, France, **5** Laboratory PHYTOX PHYSALG, Ifremer, Nantes, France

* berthelier.j@laposte.net

## Abstract

Microalgal domestication is an expanding research field that aims to multiply and accelerate the potential of microalgae for various biotechnological purposes. We investigated the stability of improved lipid traits and genetic changes of a domesticated strain of the haptophyte *Tisochrysis lutea*, TisoS2M2, previously obtained by a mutation-selection improvement program. After 7 years of maintenance, TisoS2M2 still displayed improved lipid traits compared with the native strain, demonstrating that a mutation-selection improvement program is suitable for obtaining a domesticated strain with stable, improved phenotype over time. We identified specific genetic variations between the native and domesticated strains and focused on the dynamics of transposable elements (TEs). DNA transposons mainly caused specific TE indels of the domesticated strain TisoS2M2, and some specific TE indels may have impacted genes associated to the neutral lipid pathway. We revealed transposition events for TEs in *T. lutea* and discussed on the potential role of the improvement program on their activity.

## 1. Background

Microalgae are regarded as a promising resource to tackle the increase in global population and depletion of natural resources [1] and have been studied for multiple applications, such as health, feedstock, cosmetics, bioremediation, and biofuels [2, 3]. Indeed, they constitute a sink of valuable molecules, and their short generation times make them interesting candidates for industrial purposes [2, 3]. The haptophyte *Tisochrysis lutea* was originally isolated from tropical seawater (Tahiti, French Polynesia). This microalga is currently used in aquaculture and is widely studied in the ecophysiology and biotechnology fields [4–9]. Its high content of neutral lipids makes it a potential source of biofuels [10–13]. *T. lutea* is also considered a promising feedstock for health compounds because this microalga accumulates high contents of xanthophylls (such as

**Funding:** This work was fund by the French Research Institute for Exploitation of the Sea (IFREMER), the French Region of Pays de la Loire with the Atlantic Microalgae program, the DynAlgue ANR JCJC, and LANCOM Pari Scientifique 2020 Pays de La Loire. The funders had no role in study design, data collection and analysis, decision to publish, or preparation of the manuscript.

**Competing interests:** The authors have declared that no competing interests exist.

fucoxanthin) and polyunsaturated fatty acid (PUFA) (such as docosahexaenoic acid, DHA) that play roles in the health and development of aquatic organisms. Fucoxanthin is associated with numerous health benefits, including antioxidant and anti-inflammatory effects. DHA improves the cardiovascular condition and fetal development, making this microalga potentially attractive to be used in various medicinal applications [14, 15]. Despite the promising use of microalgae such as *T. lutea* in biotechnological applications, several factors must be overcome before microalgal resources become economically viable on an industrial scale [16]. One current issue pointed out by the scientific community is the lack of knowledge on the stability of the improved traits of domesticated strains over time [16–18]. While the domestication of crops started thousands of years ago through selection and breeding for thousands of generations [19], the domestication of microalgae has a history of only two decades. Unlike crops, the sexual reproduction of microalgae is mostly uncontrolled, and their domestication is instead performed by improvement programs, the most common of which are genetic engineering [20, 21], controlled evolution [22, 23], selection [24, 25], and mutation-selection [10, 26]. While domesticated strains produced right after these improvement programs demonstrate improved traits, the stability of the improved traits over time has, to our knowledge, never been investigated after years [17, 18]. Indeed, most studies assumed that phenotypic traits measured in the laboratory are stable over time. They underestimate the possibility that the domesticated strains could turn back to the wildtype phenotype or have a drift of the improved traits [17, 18]. Such an evaluation is of major importance in the context of industrial applications [16]. In addition, while they encode for the genetic novelties giving rise to the improved domesticated traits, genetic changes in the genome of domesticated strains have been few investigated [22, 27–32]. For instance, previous studies on the genome of domesticated strains of *Chlamydomonas reinhardtii* identified new single-nucleotide polymorphisms (SNPs) and DNA deletions/insertions (indels) compared with the native strain [22, 29]. For the haptophyte *T. lutea*, we have previously investigated the genetic changes in two clonal domesticated strains compared with the native strain. Among the indels, DNA sequence similarity analysis suggested the involvement of transposable elements (TEs) [11]. TEs are DNA sequences found in eukaryotic and prokaryotic genomes that can move and spread [33, 34]. In eukaryotes, TEs are commonly grouped into two classes according to their mechanism of mobility: class I retrotransposons move by a copy-paste mechanism via RNA intermediaries; class II DNA transposons move by cut-paste mechanisms [34]. Although TE indels are mostly harmful or neutral to the host, they can also participate in the rise of evolved phenotypic traits [35, 36]. For plant crops, TEs have been found to promote various domesticated traits [35, 37, 38]. Because of the major role of TEs in the domestication of organisms, we previously performed an accurate annotation of TEs in *T. lutea*. We identified two DNA transposon families, hAT/*Ace* and Tc1-Mariner/*Luffy*, which can be expressed and get their transposase processed by the cell [39]. We suggested that these DNA transposon families may participate in the genome dynamics of the *T. lutea*.

In the present study, we evaluated the phenotypic stability over time of the domesticated *T. lutea* strain TisoS2M2, previously obtained by a mutation-selection program, which has a higher total fatty acid (TFA) content compared to the native strain TisoArg [10]. We next investigated specific genetic changes in the domesticated strain compared with the native strain by focusing on the dynamics of TEs and their potential role in the improved phenotype of TisoS2M2.

## 2. Results

### 2.1 TisoS2M2 has stable and improved lipids trait over time

The domesticated strain TisoS2M2 had previously been obtained by submitting the native TisoArg strain to a mutation-selection improvement program [10] (Fig 1A). This experiment

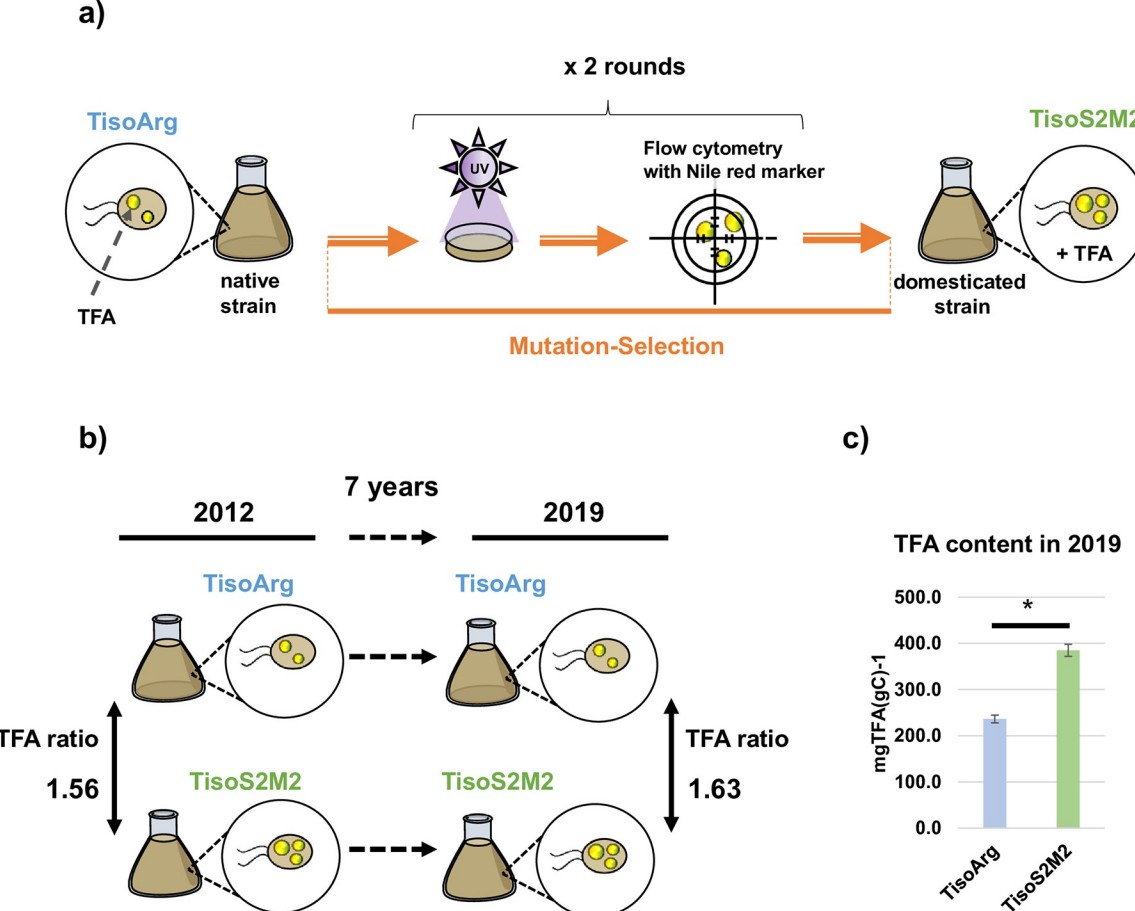

**Fig 1. Strategy of the improvement program and comparison of the total fatty acid (TFA) content between the domesticated strain TisoS2M2 strain and the native strain TisoArg.** a) Illustration of the strategy of the improvement program used in 2012 [10] to obtain TisoS2M2. b) Comparison of TFA ratio obtained in 2012 [10] and 2019 (this study) for TisoArg and TisoS2M2. c) Comparison of the TFA content of TisoArg and TisoS2M2 measured in 2019. Bars represent the means of two biological replicates ± SD. *, p < 0.05 by t-test.

was conducted in 2012 and the TFA content of TisoS2M2 was 1.56 times higher than that of TisoArg [10] (Fig 1B). Here, the TFA content of both strains was again measured in 2019. A mean of 236.4±8.4 mgTFA(gC)$^{-1}$ for the wildtype TisoArg and 385.1±13.1 mgTFA(gC)$^{-1}$ for the domesticated TisoS2M2 were found, which were significantly different (Fig 1C; S1 Data). In 2019, the TFA content of TisoS2M2 was 1.63 times higher than TisoArg (Fig 1B; S1 Data). Interestingly, the proportions of monounsaturated fatty acid (MUFA), saturated fatty acids (SFA) and PUFA among TFA were similar between both strains (S1 Data; S1 Fig). Among PUFA, we found that TisoS2M2 had a significantly higher content of ΣΩ3 and ΣΩ6 compared with TisoArg. We measured a ΣΩ3 and ΣΩ6 content of 13.6±0.6 mgΣΩ3(gC)$^{-1}$ and 11.0±0.3 mgΣΩ6(gC)$^{-1}$ in TisoArg, but 24.9±0.2 mgΣΩ3(gC)$^{-1}$ and 19.3±0.6 mgΣΩ6(gC)$^{-1}$ in TisoS2M2 (S1 Data; S1B Fig). The contents in ΣΩ3 and ΣΩ6 were 1.83 and 1.76 times higher in TisoS2M2 compared with the native TisoArg (S1 Data). Similar growth rates were found for both strains in 2019 with 0.37±0.03 µmax (day−1) and 0.35±0.08 µmax (day−1) for TisoArg and TisoS2M2, respectively, under the chosen growing condition (S1 Data; S1C Fig). In conclusion, the TFA content of TisoS2M2 continued to be superior to TisoArg.

## 2.2 Prediction of specific polymorphisms in the native and domesticated strains

In addition to the phenotypic analyses, we investigated specific genetic variants in the native and domesticated strains. The genomes of TisoArg and TisoS2M2 were sequenced with Illumina technology in 2018. Overall, the paired-end reads of each strain covered 99% of the reference genome assembly of *T. lutea* CCAP927/14. Read depths of 58.3× and 59.0× were obtained for TisoArg and TisoS2M2, respectively. First, the detection of SNPs in TisoArg and TisoS2M2 was performed. We found 52,122 SNPs in TisoArg and 47,300 SNPs in TisoS2M2. Among them, 33,671 SNPs were common to both strains, while 18,451 and 13,629 SNPs were specific to TisoArg and TisoS2M2, respectively (Fig 2A, S2 Data).

Next, we investigated specific indels in TisoArg and TisoS2M2 and focused on the ones caused by TEs, because we revealed in a previous study that the *T. lutea* genome contains autonomous TE families encoding for transposase [39]. We designed a bioinformatic pipeline to predict specific TE indels in TisoArg and TisoS2M2 genomes (S3 Data). Combining the indel predictions of the tools Pindel and Breakdancer, 57 and 53 deletions were overlapping and considered to be shared [Fig 2B; S4 Data]. However, we detected 20 TisoArg-specific deletions and 34 TisoS2M2-specific deletions. Focusing on TisoS2M2-specific predictions, we identified two high confidence deletion events detected by both tools. The first was a deletion of 862 bp located at contig 93 (S4 and S5 Data). The edges of the predicted deletion sequence display terminal inverted repeats (TIRs) identical to those of the DNA transposon family *Ace*

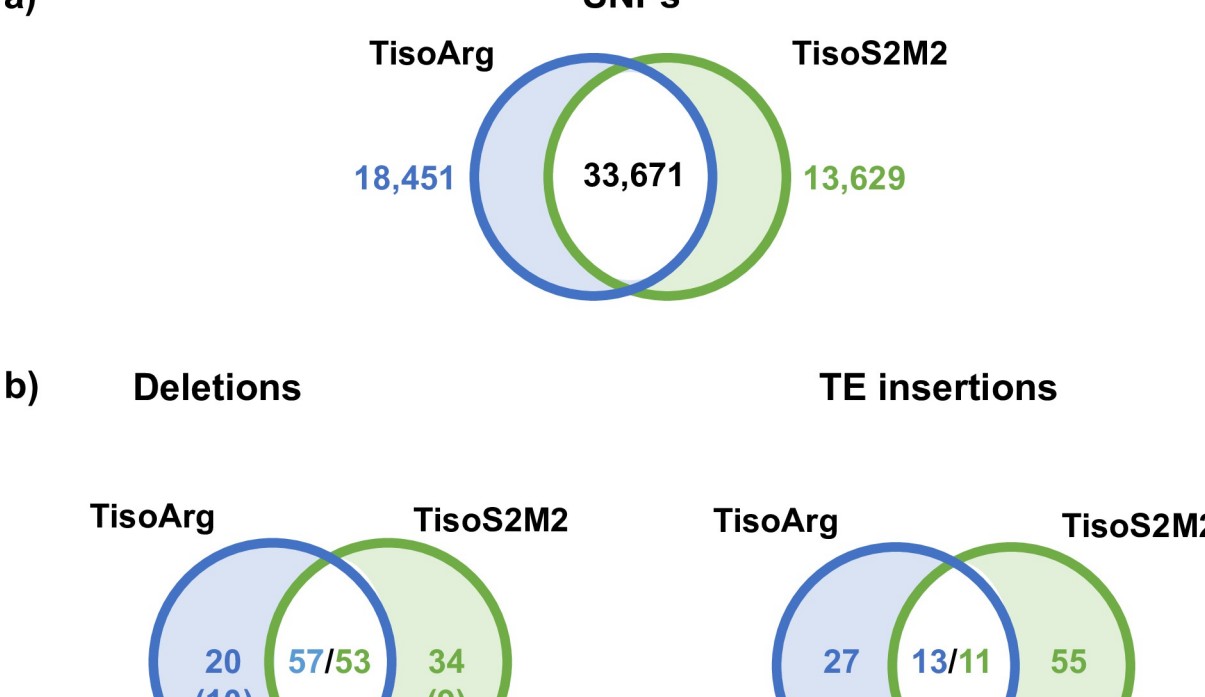

**Fig 2.** a) Shared and specific SNPs predicted for the wildtype TisoArg and the domesticated strain TisoS2M2 using Illumina reads. b) Shared and specific deletions (left) and insertions (right) predicted for the wildtype TisoArg and the domesticated strain TisoS2M2 using Illumina reads. The number of TEs associated with the deletion events is given in parentheses.

[39] (S5 Data). However, the short length of the sequence indicated that it was caused by a potential non-autonomous element. The second deletion event of 1,100 bp was predicted at contig 142. The deleted sequence had no similarity to any previously detected TEs of *T. lutea*. However, we detected TIRs belonging to a potential 531-bp TE sequence [39] (S4 and S5 Data). From the TIR sequences, we identified several copies of a new TE family in the reference genome of *T. lutea* (S6 Data), which we named *Shanks* (see 2.4) and produced an annotation of TE copies belonging to this family (S6 Data). Using this improved TE library, we found that 10 TisoArg-specific deletions and 9 TisoS2M2-specific deletions corresponded to TEs described in *T. lutea* (S4 Data). We next investigated specific TE insertion events in TisoArg and TisoS2M2. Considering the new TE family *Shanks*, we detected 26 TisoArg-specific and 55 TisoS2M2-specific TE insertions (Fig 2B; S4 Data). We found that 13 and 11 predictions of specific TE insertions in TisoArg and TisoS2M2 were overlapping. These predictions were considered to be shared (Fig 2B; S4 Data).

## 2.3 Long read sequencing analysis confirmed specific TE indels predicted by Illumina sequencing

Because numerous specific TE indels were predicted in TisoArg and TisoS2M2, we also sequenced in 2018 the genomes of TisoArg and TisoS2M2 with the Oxford Nanopore Technology (ONT). This technology produces long read sequencing, making possible to span the entire length of TEs and flanking genomic regions [40]. Our objectives were to validate TE indel events predicted by the Illumina sequencing strategy, next to confirm the TE families predicted to be associated and to discover the structure of the involved sequences (autonomous or non-autonomous TEs). Read depths of 4.1× and 7.9× were obtained for TisoArg (37,516 reads) and TisoS2M2 (91,449 reads), respectively. The tool Sniffles predicted 166 and 240 deletions in TisoArg and TisoS2M2. Among them, 90 deletions were shared between TisoArg and TisoS2M2, and none contradicted the predictions obtained with Illumina reads (S4 Data). Sniffles did not retrieve any of the specific TE deletions predicted with Illumina reads in TisoArg but confirmed the existence of 2 out of the 9 TisoS2M2-specific TE deletions detected by Illumina reads (Fig 3A; S4 Data). Sniffles predictions supported the TE deletions caused by a Tc1-Mariner/*Luffy* at the contig 97 and a hAT/*Ace* at the contig 93 (S2A Fig; S4 Data). Focusing on the second one, we retrieved 5 reads with the TE sequence, 1 read without the TE sequence and 2 reads without the TE sequence but bearing a footprint of a target site duplication (TSD) (S2 Fig; S7 Data).

Concerning insertion events, Sniffles detected 311 and 439 insertions in TisoArg and TisoS2M2, respectively (S4 Data). Among those, 19 were shared between strains, and none contradicted Mobster predictions (S4 Data). Sniffles confirmed 16 out of the 27 TisoArg-specific TE insertions and 36 out of the 55 TisoS2M2-specific TE insertions predicted by Mobster with Illumina reads (Fig 3; S4 Data). Long reads of TisoArg confirmed the presence of 12 hAT/*Ace*, 3 Tc1-Mariner/*Luffy*, and 1 Harbinger7 (Fig 3A; S4 Data). The lengths of the insertions predicted for these elements were superior to 2,500 bp, indicating the involvement of autonomous TEs (S3 Fig and S4 Data). For TisoS2M2, long reads confirmed the presence of 31 hAT/*Ace*, two hAT/*Shanks*, two Tc1-Mariner/*Luffy*, and one Harbinger7 (Fig 3B; S4 Data). The lengths of the insertions suggested the involvement of autonomous TEs for the hAT/*Ace*, Tc1-Mariner/*Luffy* and Harbinger7. Among the hAT/*Ace* insertion events, one was two times longer than a regular autonomous *Ace* and corresponded to a chimeric TE sequence composed of two distinct autonomous *Ace* elements (S3 Fig). On the other hand, insertions associated with the hAT/*Shanks* family were two times shorter than the described length of autonomous elements, suggesting the involvement of non-autonomous elements (See 2.4, S3 Fig; S4 Data).

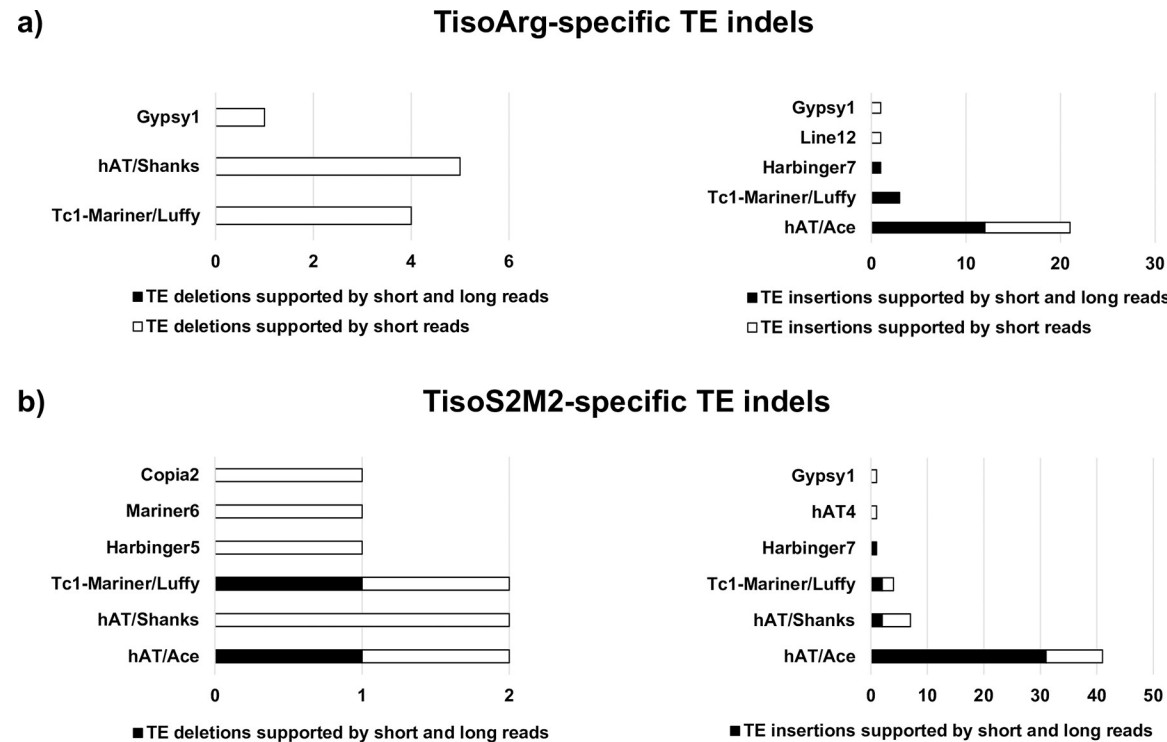

**Fig 3.** a) Proportion of TE families involved in a) specific TE indels of TisoArg supported by Illumina short reads and ONT long reads b) specific TE indels of TisoS2M2 supported by Illumina short reads and ONT long reads.

## 2.4 Characterization of TE families to improve the TE annotation of *T. lutea*

Most of the specific TE indels identified in TisoS2M2 were related to three DNA transposon families: *Ace*, *Shanks*, and *Luffy*. Because of the high involvement of these TE families, we characterized their autonomous elements and annotated potential non-autonomous TEs to improve the current TE annotation of *T. lutea* [39].

The *Luffy* family had been previously identified in the reference genome of *T. lutea* with 20 potential autonomous copies [39]. Here, we noted that the autonomous elements of this family have a DNA sequence length of approximate2,567 bp, terminal sequences of 14 bp, and are bordered by "TA" TSD motifs, which are characteristic of the Tc1-Mariner superfamily [34]. This family has a single open reading frame (ORF) encoding for a peptide of 463 amino acids (aa; S4 Fig). No domain predictions were obtained for the ORF with Pfam, but HHpred detected a transposase domain belonging to the Tc1-Mariner superfamily. We searched for potential non-autonomous elements of this TE family (bearing intact TIRs) using an autonomous referent sequence, but none were identified.

The *Ace* family is composed of three autonomous elements with a length of approximate 2,590 bp, a terminal sequence of 12 bp, and are bordered by TSD motifs of 8 bp (S4 Fig; S6 Data). They have a single ORF encoding for a peptide of 713 aa. Pfam detected two protein families (PF04937 and PF05699) related to the TIR/*hAT* superfamily. We generated an accurate annotation of potential non-autonomous TEs for this TE family and found 92 potential non-autonomous TEs, with lengths ranging from 244 to 1,632 bp. Among them, 55 elements

displayed potential intact TIRs with TSD motifs, and 37 had different TIRs, TSD motifs, or both (S5 Data).

Autonomous *Shanks* elements have a DNA sequence length of approximate 6,768 bp, non-identical TIRs of 26 bp, and are bordered by TSD motifs of 7 bp (S6 Data). They bear two ORFs encoding for enzymes of approximate 793 aa (ORF1) and 580 aa (ORF2; S4 Fig). However, no predictions were obtained for either ORF with Pfam or HHpred. During the course of this study, the TE databank Repbase classified this TE in a small subgroup of transposons TIR/hATv, composed of 16 elements, belonging to three members of the Haptophyta (*T. lutea*, *Chrysochromulina* sp. and *Emiliania huxleyi*). We annotated 1,429 copies in the reference *T. lutea* genome for the *Shanks* family. Among them, we identified 30 potential autonomous TE copies and detected 80 potential non-autonomous TEs belonging to this family. The 80 non-autonomous elements have lengths ranging from 486 to 3,050 bp, and 66 have intact TSD motifs (S6 Data). The new annotations obtained for *Ace* (92 non-autonomous TE) and *Shanks* (1,429 copies) were added to the previous TE annotation of *T. lutea* [39], constituting the second version of the TE annotation of *T. lutea* (see Material and Methods).

## 2.5 Impact of SNPs and TE indels on genes

Assuming that the improvement program induced specific genetic variants between TisoArg and TisoS2M2, we searched for SNPs or TE indels that could have impacted genes and be causal in the improved phenotype of the domesticated strain. For TisoArg, 804 of the specific SNPs were intragenic, and 385 were predicted to impact the coding sequence by promoting missense variant or change in stop codon (S2 Data). Concerning TisoS2M2, 745 specific SNPs were intragenic, and 330 were predicted to impact the coding sequence (S2 Data). We next focused on the impacted genes that are associated with lipid or catabolic metabolism using Gene Ontology (GO) terms. We retrieved 11 and 9 candidate genes in TisoArg and TisoS2M2, respectively (S2 Data). We searched for homologous genes in the literature, but none have been found to regulate the TFA content of any organisms (S2 Data).

Next, we investigated the potential impact of TE indels located within or close to genes. For instance, de novo TE insertions can drive change in the host phenotype by affecting genes [35, 36, 41]. In TisoArg, 37 TE indels were located within or close (±1,000 bp) to 57 genes. For TisoS2M2, 35 TE indels were predicted to impact 41 genes (S8 Data). Using GO terms, we found 3 candidate genes in TisoArg that were associated with catabolic or lipid metabolisms. However, none had a predicted function that could be related to the regulation of the TFA content. In TisoS2M2, we found nine candidate genes potentially impacted by specific TE indels. Interestingly, two gene candidates had homologous genes known to be able to regulate the TFA content. First, an insertion of a hAT/*Shanks* is located downstream (-511 bp) to the gene Tiso_v2_10131 predicted to encode a Cyclin-dependent kinase (S5 Fig; Table 1; S8 Data). Second, a hAT/*Shanks* deletion event is located inside the gene Tiso_v2_15006 predicted to encode for a Long-chain-enoyl-CoA hydratase and/or downstream (-50 bp) to the gene Tiso_v2_15005 predicted to encode for a class 3 lipase (S5 Fig; Table 1; S8 Data). The gene predicted to encode for the long-chain-enoyl-CoA hydratase has a length of 37,896 bp and seems to correspond to a false positive gene prediction. This gene prediction was not discussed further. Long read alignments made possible to visualize the deletion and insertion events in the TisoS2M2 dataset (S5 Fig).

## 3. Discussion

This study evaluated the phenotypic stability and genetic changes in the lipid-enhanced domesticated strain TisoS2M2 of *T. lutea*. This domesticated strain was obtained during a

**Table 1. Specific TE indels of TisoS2M2 located close to or within genes involved in lipid or catabolic metabolism.** The frequencies of the insertion and deletion events were measured by Mobster and Pindel, respectively.

| Indel ID | Contig | Start | End | TE family | Gene ID | Predicted enzyme | GO term | TE location | Frequency (%) |
|---|---|---|---|---|---|---|---|---|---|
| Pindel-TisoS2M2-19 | 142 | 46892 | 48002 | hAT/*Shanks* | Tiso_v2_15006 (potential false positive) | Long-chain-enoyl-CoA hydratase | lipid | Inside gene | 17 |
| Pindel-TisoS2M2-19 | 142 | 46892 | 48002 | hAT/*Shanks* | Tiso_v2_15005 | Lipase Class 3 | Lipid and catabolic | Downstream gene (+50 bp) | 17 |
| Mobster-TisoS2M2-47 | 016 | 260041 | 260041 | hAT/*Shanks* | Tiso_v2_13322 | Arylsulfatase | Lipid and catabolic | Inside gene | 23 |
| Mobster-TisoS2M2-272 | 086 | 172943 | 172943 | hAT/*Shanks* | Tiso_v2_10131 | Cyclin-dependent kinase A-1 | Lipid and catabolic | Upstream gene (-511 bp) | 28 |
| Mobster-TisoS2M2-747 | 189 | 830328 | 830328 | hAT/*Ace* | Tiso_v2_06146 | 60S ribosomal L10-2 | catabolic | Upstream gene (-324 pb) | 38 |
| Mobster-TisoS2M2-476 | 128 | 377408 | 377408 | hAT/*Ace* | Tiso_v2_11435 | Mitogen-activated kinase 5 Short | Lipid and catabolic | Upstream gene (-709) | 38 |
| Mobster-TisoS2M2-138 & 139 | 043 | 312781 | 312781 | hAT/*Shanks* | Tiso_v2_09258 | RAC family serine threonine- kinase homolog | Lipid and catabolic | Downstream gene (+392) | 10 |
| Mobster-TisoS2M2-138 | 043 | 312776 | 312776 | hAT/*Shanks* | Tiso_v2_09258 | RAC family serine threonine- kinase homolog | Lipid and catabolic | Downstream gene (+397) | 30 |
| Mobster-TisoS2M2-699 | 179 | 341961 | 341961 | Tc1-Mariner/*Luffy* | Tiso_v2_09731 | Cystathionine gamma-synthase | Catabolic | Upstream gene (-27) | 37 |
| Mobster-TisoS2M2-351 | 097 | 1811743 | 1811743 | hAT/*Ace* | Tiso_v2_02964 | Unknown | Catabolic | Downstream gene (+853 pb) | 27 |

previous study from a native Wt strain TisoArg through a UV-based mutation-selection program [10]. UV radiation is a robust and inexpensive method to generate random mutations and obtain a pool of mutants without previous knowledge of the genetics and metabolisms of the studied microalgae [18]. However, random mutations can be deleterious for cell fitness, and the obtained domesticated strain with a phenotype of interest could occasionally revert to wildtype over time [18, 42]. While the stability of improved phenotypic traits is essential in an industrial context, studies investigating the phenotypic stability of domesticated strains are lacking [18]. Here, we confirmed that seven years after a UV-based improvement program, the domesticated strain TisoS2M2 had a stable, lipid-enhanced trait over time. By comparing the TFA content of the domesticated and native strains in 2019, we revealed that TisoS2M2 had 1.63 times higher TFA content than the native. This ratio is very similar to the ratio previously measured in 2012 (ratio of 1.56) [10]. In addition, a similar proportion of PUFA, SFA and MUFA was found among the TFA content of TisoArg and TisoS2M2 (S1 Data and S1A Fig). The increase in TFA content in TisoS2M2 is therefore caused by a global increase of the fatty acid groups, which is stable over time. The growth rates of TisoS2M2 and TisoArg were very similar and not statistically different in 2019, as also reported in 2012 [10]. Our results confirmed that the improvement program of Bougaran et al. is suitable for the creation of domesticated strains with a stable and improved TFA trait over time [10]. For over two decades, forward genetics strategies comprising random mutagenesis have been used with microalgal strain. Improvement of microalgae was conducted in response to the growing demand for biomolecules of interest such as PUFAs and antioxidant molecules, biomass productivity [20, 43–46], photosynthetic efficiency [47], pesticide tolerance [48], global demand for sustainable biofuels [10, 47, 49–62]. It is, therefore, crucial to control that the improved phenotypic traits of domesticated strains are stable over time [63].

The genetic characterization of domesticated strains is also important to identify the genetic mutation(s) that caused the improved phenotype and to control its persistence over time. Few studies investigated genetic differences between native and domesticated strains, and some identified the causal genetic mutation that gave rise to the improved phenotype [27, 28, 30–32]. A previous study used a reference genome assembly and removed shared background variants between the wildtype strain and mutants to identify the causal mutations [27]. We used a similar method to characterize the genetic changes (SNPs and TE indels) between TisoArg and TisoS2M2. While a part of genetic variants could have spontaneously appeared during the maintenance period, we assumed that the UV-based improvement program caused most of the specific SNPs and TE indels in TisoS2M2.

UV is known to generate random mutations such as SNPs in algal genome [27, 64, 65], and this type of mutation can be causal in the improved phenotype of a domesticated microalgae [27]. We identified thousands of specific SNPs in TisoArg (18,451 SNPs) and TisoS2M2 (13,629 SNPs), and hundreds were predicted to have a potential impact on candidate genes associated with "lipid" or "catabolic" GO terms. However, we found no homologous genes that have been previously described to regulate the TFA content in the literature. This analysis suggested that the improvement program induced changes at the single nucleotide level in TisoS2M2, leading to specific SNPs predictions in both strains. According to our analysis, SNPs did not seem to be the causal type of mutation associated with the lipid-enhanced phenotype of TisoS2M2. However, the fact that predictions of SNPs and their impacts on genes are challenging, and that the knowledge about gene functions of *T. lutea* is limited, may have prevented the detection of causal mutation(s).

TEs are known to be major actors in the genome evolution of organisms and can give rise to new phenotypes by impacting gene expression or function [35, 36, 41]. Because potential autonomous TEs were characterized in *T. lutea* [39], we analyzed the dynamics of TEs in the genome of TisoS2M2. Previous works combining Illumina and ONT sequencing technologies were performed to identify indels [31, 66].

Using this strategy, we found numerous specific TE indels in TisoArg and TisoS2M2. About half of the predictions obtained by the tools using Illumina sequencing were supported by ONT long reads (Fig 3). Most of the specific TE indels were associated with three DNA transposons families indicating their ability to transpose in the genome of *T. lutea*: the TE families hAT/*Ace* and Tc1-Mariner/*Luffy*, which had previously been reported as expressed in the reference genome of *T. lutea* [11], as well as the hAT/*Shanks* family, newly identified in the present study. Most of the TE indels spanned by ONT long reads were found to be caused by autonomous TEs. However, we also revealed that several TE indels, associated with hAT/*Shanks* families, were caused by potential non-autonomous TEs. We annotated many potential non-autonomous elements of the hAT/*Ace* and *Shanks* families in the reference genome of *T. lutea*, suggesting that the *T. lutea* genome hosts numerous non-autonomous TEs belonging to *Ace* and *Shanks* famil*es* that could be mobilized. Long read alignments made also possible to pinpoint a locus with heterozygous TE indel events in TisoS2M2 (S2 Fig and S7 Data). The life cycle of *T. lutea* is still unknown, but is potentially haploid-diploid as described for other haptophytes [67]. The life cycle of *T. lutea* is uncontrolled at laboratory conditions and could lead to the presence of haploids and diploid cells in the algal population.

The finding of specific TEs in both the native TisoArg and the domesticated TisoS2M2 was unexpected. We assumed that the UV-based steps of the improvement program previously conducted in 2012 could have promoted TE mobility leading to specific TE indels in TisoS2M2 [10] (Fig 1A). Conversely, the selection effects of the improvement program could have discard haplotypes of the native TisoArg with specific TE loci, which were not selected in the domesticated TisoS2M2 [10, 68] (Fig 1A). Another hypothesis would be that specific TE indels in

TisoArg and TisoS2M2, or a part, could have appeared during the years of maintenance. TE mobility events were recently reported in a strain of *C. reinhardtii* maintained at laboratory condition [69]. However, almost two times more specific TE indels were found in TisoS2M2 compared to TisoArg, reinforcing the hypothesis that the improvement program had an impact on the mobility of TEs in TisoS2M2.

In microalgal field, previous studies found that TE indels can affect genes, causing phenotypic changes [70–73]. For instance, a previous study identified that an insertion of a TE upstream a desaturase gene caused decrease in the fatty acid contents of a *C. reinhardtii* mutant [74]. Assuming that specific TE indels in TisoS2M2 appeared during the improvement program in 2012 [10], we searched for TE indels that could have impacted genes known to regulate the TFA content of organisms. While no candidates were found for TisoArg, two candidate genes were identified in TisoS2M2. First, a TE insertion was found upstream of a gene encoding for a Cyclin-dependent kinase (CDK). This gene family was mainly reported to be involved in cellular processes, but a study also revealed that a CDK regulated the lipid content of *Arabidospis thaliana* [75]. Indeed, a knockout mutant of the gene CDK8 in *A. thaliana* has a significant reduction in fatty acid content. Conversely, a higher fatty acid content was measured in a mutant of *A. thaliana* overexpressing this gene [75]. In *T. lutea*, the TE insertion located upstream of a predicted CDK might affect its expression and could lead to an overaccumulation of the TFA content. In addition, a TE deletion was identified downstream a gene encoding for a class 3 lipase. This lipase class is known to be directly involved in the TFA degradation of plants [76], and homologous genes have also been described in microalgae [76–78]. The downregulation of class 3 lipases was notably reported to increase the TFA content of microalgal cells [77]. Nevertheless, these two TE indels are predicted with low allele frequencies (28 and 17%), and are probably not the main genetic(s) cause(s) of the lipid-enhanced phenotype of TisoS2M2. The selection of clonal strains having these TE indels would be an interesting proposition for further investigations. The causal mutation(s) of the TisoS2M2 phenotype remain to be explored.

Although this cannot be confirmed with our current data, the detection of specific TE indels in TisoS2M2 suggested that the UV induced the activity and mobility of several TE families during the improvement program. UV has been reported to trigger TE expression in crops [79, 80]. Indeed, UV caused chromatin remodeling in maize, and induced the expression of epigenetically silenced TEs [81]. A similar scenario could have happened during the improvement program of *T. lutea*. Epigenetic mechanisms are thought to play an essential role in the adaptation processes of microalgae because of fluctuating environmental conditions [82]. While epigenetics mechanisms involving DNA methylation, histone modification, or RNAi have been reported in Chlorophyte, Diatoms, and Dinoflagellate [82–86], they are still unknown for *T. lutea* and, more generally, for haptophytes. Nevertheless, knowledge about the epigenetic regulation of TE activity is still limited and appear to differs depending on the microalgal taxa [83, 87–90]. To our knowledge, no previous studies investigated the effect of UV on the activity of TEs in microalgae. However, the expression of TEs was pinpointed to be seasonally responsive to UV radiations in natural conditions for the Rhodophyta *Cyanidioschyzon* sp. [91]. An effect of UV on the TE activity of *T. lutea* during the improvement program is conceivable and would be a key question to investigate further.

## 4. Conclusion

We investigated the stability of improved lipid traits and specific genetic changes of the domesticated strain TisoS2M2, previously obtained by a mutation-selection program. After 7 years of maintenance, TisoS2M2 still conserves a higher TFA content compared to the native TisoArg,

revealing a stable, improved lipid phenotype over time. In an industrial context, evaluations of this kind are crucial to ensure the development and use of suitable domesticated strains. In addition, we characterized the genetic changes between the native and the domesticated strain by focusing on the dynamics of TEs. We predicted and confirmed numerous specific TE indels in TisoS2M2, revealing transposition events for several TE families of *T. lutea*. While candidate genes could be impacted by TE indels, the causal mutation(s) explaining the improved phenotype of TisoS2M2 is still undetermined. Our results suggest that UV may have induced the TE activity during the improvement program, which would be an important question to address in the future.

## 5. Material and methods

### 5.1 Microalgae strains and culture conditions

The wildtype *T. lutea* strain, TisoArg, was provided by the French Research Institute for Exploitation of the Sea (France) in 1998. TisoS2M2, which accumulated more TFA content compared with TisoArg, was previously obtained from an improvement mutation-selection program [10]. TisoArg and TisoS2M2 are not clonal but composed of populations of cells. For culture maintenance, these strains were grown in 200 mL flasks with natural seawater enriched with sterile Conway medium [92]. These cultures were maintained for 3 weeks, during which time they were supplied with bubbles from 0.22-mm filtered air and kept at a constant temperature of 21°C and under a continuous light irradiance of 50 μmol m$^{-2}$ s$^{-1}$.

### 5.2 Physiological characterization of the studied strains

Comparisons between the native and domesticated stains were conducted in duplicated cultures. Cultures were growth under similar conditions than ones used in the experiment of 2012 [10]. First, 300,000 cells from the mother culture were used to inoculate new culture flasks that were designed for phenotype characterization [93]. Each flask contained 300 mL natural seawater enriched with sterile Conway medium [92] with modified nitrogen content (150 μM instead of 1,500 μM). Each flask was placed in a culture control system with the following parameters: regulated temperature at 23°C; continuous light irradiance of 350 μE m$^{-2}$ s$^{-1}$; regulated pH of 7.5 with a $CO_2$ pulse. The growth of microalgae was automatically monitored by optical density measurements. The formula used to calculate the growth rate was $\mu = (Ln(X_2)-Ln(X_1))/(t_2-t_1)$. The uptake of $NO_3$ was measured twice a day as follows: (1) a 2-mL sample was collected; (2) microalgae were eliminated by filtration (0.2 μm); (3) the medium culture was analyzed by spectroscopy at 220 nm to establish the level of remaining $NO_3$ [94]. After exhaustion of the nitrogen, microalgae growth was stopped after approximately 2 days. After 1 day in stationary phase (corresponding to 6 culture days), harvesting was performed to characterize the microalgae. Each sample was filtered using two pre-combusted GF/C filters for CN and for lipid respectively (diameter, 47 mm; Whatman, Maidstone, UK). To measure the carbon content in each microalgal cell, we used a carbon elemental analyzer to test the filters, which had been placed in an oven and dried at 75°C for 24 h. The filters were deposited in bottles filled with 6 mL of Folch reagent and deep frozen (−80°C) for lipid measurement. Lipid class separation was performed by column chromatography according to the method by Soudant et al. [95], with a BPX-70 capillary column (60 m long, 0.25 mm internal diameter, 0.25 μm film thickness; SGE, Austin, TX) containing a polar stationary phase (cyanopropyl-siloxane). The upper organic phases containing fatty acid methyl esters (FAMEs) were collected and assayed by gas chromatography coupled with flame-ionization detection (GC-FID). FAME quantification was compared with the C17 internal standard (Sigma-Aldrich, St. Louis, MO) by GC-FID using a gas chromatograph

(Autosystem Gas Chromatograph; Perkin-Elmer, Waltham, MA). TFA content was calculated as the sum of SFA, PUFA and MUFA.

## 5.3 Whole genome sequencing of the *T. lutea* strains

Microalgal cultures were treated with antibiotics (A5955; Sigma-Aldrich) to avoid bacterial contamination. DNA was extracted using a phenol-chloroform method detailed in Hu et al. [96]. Sequencing was conducted with an HiSeq4000 sequencer (150 bp, paired-end; Illumina, San Diego, CA). The raw sequencing data were trimmed with TrimGalore (v0.6.2) (https://github.com/FelixKrueger/TrimGalore) to remove adapters and get a quality score higher than Q30. In addition, TisoArg and TisoS2M2 were sequenced with a MinION device (Oxford Nanopore Technologies, Oxford, UK) using the Ligation Sequencing Kit SQK-LSK109 and one flow cell R9.4 for each strain. Basecalling was performed for reads with a minimum quality score of 8 using Guppy v3.3.

## 5.4 Prediction of polymorphisms

Illumina reads were obtained independently from each strain (TisoArg and TisoS2M2). To compare the polymorphisms of each strain, reads were mapped independently on the reference genome assembly v2 of *T. lutea* (strain CCAP 927/14 [39]) using Mosaik [97] or bwa-mem [98]. SNPs were predicted with DiscoSnp++ [99, 100]. Because the reference genome assembly belongs to a different strain than the studied ones, only the predicted 0/1 and 1/1 SNPs were considered. Predicted SNPs of TisoArg and TisoS2M2 were compared with BCFtools isec [101].

## 5.5 Prediction of TE indels

Deletions were predicted with the tools Breakdancer [102] and Pindel [103], which were applied to the bwa-mem alignments of each strain. Only deletions with a minimum of 150 bp and a maximum of 15,000 bp were selected as positive candidates (see details in S3 Data). The tool PASTEC [104] was used to detect the deletions caused by TEs and predict their TE family. TE insertions were detected with the tool Mobster [105] using the Mosaik alignments obtained for each of the two strains. Mobster was used with a library of manually curated TE sequences of *T. lutea* (also containing sequences of *Shanks*). To prevent false positive events, we compared the location of the TE insertions detected by Mobster against the TE annotation v2 of *T. lutea* (see 5.6). We removed TE insertions detected by Mobster that were overlapping annotated TEs. Next, we compared the locations of TE indels detected in each strain with bedtools' intersect function [106] and selected only those specific to the domesticated strain (see details in S3 Data). Finally, The tool Sniffles was used with long reads to support TE indels predicted with Illumina sequencing dataset [107]. Sniffles predictions were not used to detect new specific TE indels because of the low read depht of the ONT long reads obtained for TisoArg or TisoS2M2. The tool PASTEC [104] was used to detect the indels caused by TEs and predict their TE family. Read alignments were visualized using the genome browser IGV [108].

## 5.6 Characterization of the TE families and new TE annotation

To annotate *Shanks* elements and non-autonomous elements of *Ace*, curated reference TE sequences were submitted to TEannot [109] implemented in PiRATE [39]. The new annotated copies were added to the previous TE annotation of *T. lutea* [39], constituting TE annotation v2. We used ORFfinder (www.ncbi.nlm.nih.gov/orffinder/), Pfam [110], HHpred [111], and

the online tool Censor [112] to characterize the autonomous TEs in the *Luffy*, *Ace*, and *Shanks* families.

## 5.7 Analysis of the genes potentially impacted by SNPs or TE indels

The gene annotation of the genome assembly v2 of *T. lutea* was used [39] to identify genes that could potentially be impacted by SNPs or TE indels. SnpEFF was used to predict the potential effect of SNPs on gene amino acid and we selected the SNPs predicted to have a "moderate" (missense variant) or "high" (changing stop codon) impact [113]. The genes located in a window ±1,000 bp from the predicted TE indel positions were retrieved using bedtools' intersect function [106]. The functional annotation of genes was used to retrieve genes with GO term associated to keywords "lipid" and "catabolic". Function of gene was also manually checked by finding the largest ORF with ORFfinder, then submitting the protein sequence to NCBI BLASTp and HMMER from The MPI Bioinformatics Toolkit [111].

## Supporting information

**S1 Fig. Additional phenotypic comparison between the native TisoArg and the domesticated strain TisoS2M2 in 2019.** a) Comparison of the proportion of SFA, MUFA and PUFA. b) Comparison of the $\Sigma\Omega3$ and $\Sigma\Omega6$ content. c) Growth curves of TisoArg and TisoS2M2. For a) and b), bars represent the means of two biological replicates ± SD. *, p < 0.05 by t-test. N.S. means Not significant. For c) Each dot represents the mean of two biological replicates ± SD.
(TIF)

**S2 Fig. Alignment of ONT long reads supporting a TE deletion event in TisoS2M2 with heterozygosity.** a) Track of the locus at Contig 93 with a predicted hAT/*Ace* deletion in TisoS2M2. The TE deletion event is visible for 3 reads. b) Illustration of the heterozygosity at the loci of the hAT/*Ace* deletion events at Contig 93 and the supporting nucleic motifs found from the long reads of TisoS2M2. The highlight of the motifs are in S7 Data.
(TIF)

**S3 Fig. Violin plot of the length distribution of specific TE insertions in TisoArg and TisoS2M2 supported by short reads and long reads.** Lengths were retrieved from Sniffles predictions in S4 Data). TE families with one predicted insertion are not shown.
(TIF)

**S4 Fig. Sequence structures of the TE families Tc1-Mariner/*Luffy*, hAT/*Ace*, and hAT/*Shanks*.**
(TIF)

**S5 Fig. Alignment of ONT long reads supporting TE indels in TisoS2M2 that may impact two candidate genes associated to lipid GO term.** a) A TE deletion is predicted close to a gene encoding for a class 3 lipase in TisoS2M2. b) A TE insertion is predicted to be closely located to a gene encoding for a CDK. The indels are highlighted in violet and the lengths of the events are indicated.
(TIF)

**S1 Data. Comparison of the lipid contents and growth rates between TisoArg and TisoS2M2 in 2019.**
(XLSX)

**S2 Data. List of shared and specific SNPs between TisoArg and TisoS2M2, and predictions of their impacts on genes.**
(XLSX)

**S3 Data. Pipeline used to predict specific TE indels in TisoArg and TisoS2M2 using Illumina sequencing data.**
(PDF)

**S4 Data. List of predictions of shared and specific indels detected by Pindel, Breakdancer, Mobster, and Sniffles in TisoArg and TisoS2M2.**
(XLSX)

**S5 Data. Predicted sequences by Pindel of the deletion events located at the contigs 93 and 142 of *T. lutea*.**
(PDF)

**S6 Data. List of annotated TEs of *Shanks* and non-autonomous TEs of *Ace*.**
(XLSX)

**S7 Data. Detailed alignment of the ONT long reads supporting the prediction of a TE deletion with heterozygosity in TisoS2M2.**
(PDF)

**S8 Data. List of genes predicted to be impacted by TE indels.**
(XLSX)

## Acknowledgments

We acknowledge the platform Genotoul GeT-PlaGe for the Illumina sequencing of *T. lutea* strains and OIST English editing service for proofreading of the manuscript.

## Author Contributions

**Conceptualization:** Jérémy Berthelier, Bruno Saint-Jean, Grégory Carrier.

**Data curation:** Jérémy Berthelier, Bruno Saint-Jean, Gaël Bougaran.

**Formal analysis:** Jérémy Berthelier, Bruno Saint-Jean, Grégory Carrier.

**Funding acquisition:** Bruno Saint-Jean, Nathalie Casse, Gaël Bougaran, Grégory Carrier.

**Investigation:** Grégory Carrier.

**Methodology:** Jérémy Berthelier, Grégory Carrier.

**Software:** Grégory Carrier.

**Supervision:** Bruno Saint-Jean, Nathalie Casse, Grégory Carrier.

**Validation:** Gaël Bougaran.

**Visualization:** Jérémy Berthelier.

**Writing – original draft:** Jérémy Berthelier.

**Writing – review & editing:** Bruno Saint-Jean, Nathalie Casse, Gaël Bougaran, Grégory Carrier.

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
