## [Decision Letter · Decision Letter 0]

28 Nov 2022

PONE-D-22-30286Phenotype stability and dynamics of transposable elements in a strain of the microalga Tisochrysis lutea with improved lipid traitsPLOS ONE

Dear Dr. Berthelier,

Thank you for submitting your manuscript to PLOS ONE. After careful consideration, we feel that it has merit but does not fully meet PLOS ONE’s publication criteria as it currently stands. Therefore, we invite you to submit a revised version of the manuscript that addresses the points raised during the review process.

We look forward to receiving your revised manuscript.

Kind regards,

Balamurugan Srinivasan

Academic Editor

PLOS ONE

Journal Requirements:

Reviewers' comments:

Reviewer's Responses to Questions

**Comments to the Author**

1. Is the manuscript technically sound, and do the data support the conclusions?

Reviewer #1: Partly

Reviewer #2: Yes

Reviewer #3: Partly

2. Has the statistical analysis been performed appropriately and rigorously? 

Reviewer #1: No

Reviewer #2: Yes

Reviewer #3: N/A

3. Have the authors made all data underlying the findings in their manuscript fully available?

Reviewer #1: No

Reviewer #2: Yes

Reviewer #3: Yes

4. Is the manuscript presented in an intelligible fashion and written in standard English?

Reviewer #1: Yes

Reviewer #2: Yes

Reviewer #3: Yes

5. Review Comments to the Author

Reviewer #1: The data presented by Berthelier and coworkers is an interesting approach in the field of applied research on microalgae. However, data and interpretations presented here are limited. In my opinion, the main issues are :

- limited data on growth rates and lipid accumulation : experiments were made as duplicates, but the deviation between duplicates is not shown. Except if duplicates are highly similar, one usually expects triplicates and statistical treatment to enforce differences between the wild type (native) and mutant. This is lacking in Fig. 1 and Suppl Fig. S1. Also, authors focus on TFA while they probably have more detailed results that could be displayed : the mat&meth describes FAMEs analyses and lipid separation. By the way, there is ambiguity : are "TFA content" (line 91), "TFA ratio" (line 92) and/or “TFA productivity” (line 98) equivalent ? How are they computed from the raw lipid analyses ?

- SNPs and indels in the native TisoArg (considered as wild-type, “WT” hereafter) as compared to reference genome are poorly commented and generally regarded as a “baseline”, while detected in a huge amount (line 114-115 : “18,451 […] SNPs were specific to TisoArg”). Are these SNPs “missed” in the mutant ? (neither SNP or indel predictors are perfect, it is a current challenge in the field, the authors may comment on that). Or when/how they were acquired in the WT (in absence of UV treatment as in S2M2)? In my opinion, this is crucial to assume that indels and SNPs in S2M2 were caused by the mutagenesis. Lines 115-119, Figure 1 and discussion should include information and interpretation on WT as well. Also, paragraph 5.5, can you explain why aligners bwa-mem and Mosaik are then used in conjunction with distinct indel/SNPs callers ? E.g. could you reinforce your detection of insertions by crossing results with Mobster on bwa aligned reads ?

- Information and interpretation derived from the Nanopore sequencing are relatively short. Methods in paragraph 5.3 could be more detailed (kits, basecalling mode, output quality and amounts,…). The contradiction between several Nanopore reads is not clear. Is it different populations ? Is this locus heterozygous (diploidy/aneuploidy in S2M2 ?) ? Has the whole region been duplicated before transposon movement? Do these reads come from populations of cells that are distinct in this region ? Also, line 265 and after are slightly ambiguous. Do you mean that out 57 indels predicted from Illumina, 55 were not covered at all by Nanopore reads ? Or that the reads were inconclusive ? Or that Illumina and Nanopore were contradictory ? This is probably crucial to your point. Have you tried to assemble de novo the WT and S2M2 genomes out of Nanopore reads (e.g using Flye, Canu, Smart de novo,…) and compare it three-way WT vs S2M2 vs reference contigs ? (You may also polish these assemblies using Illumina reads).

Overall, this work could be re-shaped (and go deeper) to focus on the discovery of Shanks transposon family and mutation rates in T. iso (including description of WT SNPs and indels), and conclude on the phenotypic stability (with the proposed details). The candidate causal mutation could be an opening because it calls for more investigation in a later publication.

Data availability:

Will the Illumina and/or Nanopore sequences be made available and how?

Minor remarks:

Some introductive elements are missing regarding S2M2: is this a “strain” (line 79) ? monoclonal ? Derived from a single cell at any step ? or is this a population (line 225) ? in the sense of being derived from the batch selection of distinct mutants ? or in the sense of genetic drift/emergence of new mutations in distinct cells during the 7 years cultivation which makes the culture closer to a population ? Reading between the lines, one understands that no causal mutation(s) has been identified for the increased lipid production in S2M2 and that this works intends to find candidate genes, whether this is the case or not, this could be mentioned. Regarding T. isochrysis in general, how large is the genome ? is it haploid/diploid ? if diploid, an idea of the level of heterozygosity ? Any hint whether S2M2 has conserved these features ?

Could you estimate how many generations represent 7 years of cultivation ? Are there estimates of the mutation rate(s) in T. iso ? Can you comment regarding your observations ?

Line 98: “furthermore” => “In conclusion”

Line 116: “hight” => “high”

Line 122: not sure of the syntax : “… enable to encode for …”

Table : line 2 and 3 have same locations, but distinct “gene ID”, can you comment ?

Line 261: this challenges the view that DNA TEs are “cut-paste”, isn’t ? Is transposition independent from the encoded transposase then? or is this observations rather due to sequencing/analyses limitations ? Since this work focuses on the transposition events, can you comment further ?

Reviewer #2: Transposable elements are important components of the eukaryotic genome. However, there are still a lot of research gaps in their genomic evolution, function and disease immunity. In this paper, the stability of lipid traits and genetic changes of TisoS2M2 were obtained by mutation-selection improvement program, and the effect of the improvement program on the TE kinetics of domesticated microalgae was determined for the first time. This study has certain innovation and reference, but some problems need to be explained.

1.Why use UV-induced induce TE mobility instead of other methods accepted by commercial and political rules?

2.The classification of transposable elements is also constantly changing, with the discovery of new element types being constantly revised and new classification scales being introduced. Methods and standards for identifying and categorizing transposable elements are also evolving, so whether there is a difference in the sequencing results from a few years ago.

3.The article mentioned that some TE indels may affect genes involved in the neutral lipid pathway, but it did not verify which genes are changed in expression, whether it had a positive effect or a negative effect.

4.In the "Results Section", the number 2.1 is followed by the number 2.3, and the number 2.2 is missing. Please correct it.

5.Tisochrysis lutea is not described in detail in Background, and what are the special features that led the author to study this type of algae

6.The reference format is not uniform, and the case is not standardized.

Reviewer #3: This study builds on previous work by the research group, describing changes in transposable elements in a strain that was improved to have improved lipid content through mutagenesis. The authors use the framework of domestication for the paper, which is appropriate and timely. The work is a first step to understand the mechanisms by which mutagenesis-selection regimes affect traits and/or trait stability. However, there are a few flaws that need attention. First (and glaringly), with the exception of their own work (reference 32), there is no discussion of previous work on transposable elements in algae. A decent amount of work has been done in Chlamydomonas and there is also relevant literature that can be pulled from the macroalgal world. Second, the paper seems to have been “stretched” with previous work. That is, methods and results of the previous work already published area included here. With a generous read, it is not necessary to do this and a shorter publication that explicitly states they are building off the former would still add to the body of literature. Third, the paper falls short of describing mechanisms in section 2.6. Finally, the paper needs restructuring (results are in the discussion, discussion is in the results). Specific comments are below.

Major comments

It seems all of the results in section 2.1 were published elsewhere. If this is the case, this content should be part of the materials and methods as the current study builds on the former. The results section should start with 2.2 (mislabeled 2.3 Prediction of polymorphism….) as that is the new work. I would also more explicitly state throughout the manuscript that this work builds on former publications.

Figure 1b is fine but not necessary, would consider whether this is worth including.

The second results section 2.2 (mislabeled 2.3) also starts with previously published work. This could also be moved to the methods section.

Lines 154-157: It should first be stated that 57 indels were identified, as later text refers to this number but it is easily glossed over here.

More references are needed for the sentence in line 213.

Section 2.6 could hold the major crux of the paper, but there is very little content within it. The authors state the 31 insertions and the two deletions are located within or close to genes, and that nine were close to genes involved in lipid or catabolic metabolism. But only one deletion event is described. What about the other eight? There is some information on these in Table 1. Can these results be discussed further?

The authors bring up trait evolution. Although not related to lipid metabolism, I am curious about other traits that changed between the time of mutagenesis and present. Was there drift in other traits? Other trade-offs? There is room here for a broader analysis.

Reading through the results, I thought the paper was structured as an integrated results and discussion section. As they are separate, there is content that should be pulled from the results and placed in the discussion (e.g., lines 165-166; 168-169; lines 222-226). Similarly, the discussion section reads like a results section and is repetitive in many sections. For example, there is little to no new content in the first two pages of the discussion section. Both sections need to be considerably restructured.

Line 285, the discussion of the mutation-selection program, seems an appropriate place to start the discussion – and could be a quite interesting discussion point. Yet, content there is lacking. The authors reference the use of UV mutagenesis in plants (one reference), but not algae. I would integrate the relevant algal literature. I might also expand the discussion on epigenetics a bit.

The Conclusion includes prior work. Needs to be altered to consider only current work.

In the methods, Sections 5.1., 5.2, and 5.3 appear to be the previous work.

Lines 377 and 378: why were two programs used to map the assemblies? Could this have led to different artifacts?

Minor comments

The manuscript could use another read though to clean up grammar, sentence structure and flow. These stood out to me:

Lines 58-60: reference of “it” is misplaced, rephrase

Line 64: the native strain

Line 67: which should be that, or comma should be added

Line 68: change they to transposable elements

Line 70: While should be although

Line 82: compared to

Line 116: typo in hight

Line 167: compared

Line 168: this result suggests

Line 179: delete around, here and in other places; or replace with approximately if the bp number is approximate (it seems the number is precise, so may also change bp number)

Line 199: add comma after study

Line 200: add comma after TIR/hATv

6. PLOS authors have the option to publish the peer review history of their article (what does this mean?). If published, this will include your full peer review and any attached files.

Reviewer #1: **Yes: **Frederic Chaux

Reviewer #2: No

Reviewer #3: No

---

## [Author Response · Author response to Decision Letter 0]

1 Mar 2023

We have provided a PDF file with all the responds to reviewers (and tables). Below is the text version as requested:

----Responses to the reviewer’s comments----

First of all, we would like to thank the reviewers for their constructive and valuable comments on our manuscript. Below, we indicated the major changes made in the revised manuscript, and further provided point-by-point responses to the reviewers’ comments.

Major changes and additional results

- Datasets of the total fatty acid (TFA) and optical density (OD) were provided in Data S1. 

The growth rates were revised. We wrongly reported 0.031 and 0.038 µmax (day−1) for TisoArg and TisoS2M2, respectively. These values were obviously very low for the species. In the revised manuscript and Figure S1c, we are now providing the corrected OD curves and the corrected growth rates of TisoArg and TisoS2M2. We now report a growth rate of 0.37 and 0.35 µmax (day−1) for TisoArg and TisoS2M2, respectively. This is consistent and very similar to the previous finding reported by Bougaran et al. 2012, where the growth rates were of 0.40 and 0.38 µmax (day−1) for TisoArg and TisoS2M2, respectively. The formula used to measure the growth rate was added in the Material and Methods.

We noticed that the TFA ratio previously reported between TisoArg and TisoS2M2 had to be corrected due to a bad table sheet manipulation. We are reporting a corrected ratio of 1.63 between TisoArg and TisoS2M2 in 2019.

- Detections of SNPs and INDELs of the Wt strains (called TisoArg) were performed as requested. Manuscripts, Figures, Data S3 and Data S4 were revised. 

- The detection of INDELs using long reads (from Oxford Nanopore Technology sequencing) was revised. We re-performed this analysis using a different tool named “Sniffles”. This tool, “Sniffles” outperformed the tool “LorTE”, that we previously used in the first manuscript version. In the previous manuscript version, “LorTE” retrieved 2 out of 57 INDELs predicted by tools using Illumina sequencing data. We now provide a new analysis using the tool “Sniffles” which retrieved and confirmed about half of the TE INDELs predicted by tools using Illumina sequencing data. The manuscript was globally revised with new results. 

Previous results and discussions made from “LorTE” analysis were removed in the revised manuscript.

- In the revised manuscript, the criteria to predict deletion events were changed. We now combined deletion predictions obtained by Pindel and Breakdancer (revised Fig. 2 and revised Data S3). Next, because we focused on deletion events caused by TEs, we discarded deletions predicted by Illumina sequencing with a length inferior to 150 bp and superior to 15,000 bp. 

These new criteria allowed retrieving and confirming more deletion events found by Illumina and nanopore sequencing (Fig. 3). These changes were detailed in Materials and Methods.

- Additional analyses and new Figures were provided (Fig. 1c , Fig. 2, Fig. 3, Fig. S1, Fig. S2a, Fig. S3, Fig. S5) as requested by reviewers.

- Results and discussion parts were re-shaped as requested by reviewers.

----Point-by-point response to the reviewers’ comments----

Reviewer #1: The data presented by Berthelier and coworkers is an interesting approach in the field of applied research on microalgae. However, data and interpretations presented here are limited. In my opinion, the main issues are :

- limited data on growth rates and lipid accumulation: experiments were made as duplicates, but the deviation between duplicates is not shown. Except if duplicates are highly similar, one usually expects triplicates and statistical treatment to enforce differences between the wild type (native) and mutant. This is lacking in Fig. 1 and Suppl Fig. S1. 

Our response: We agree with reviewer’s comments. The data of the optical density (OD) used to measure the growth rate and the data used to measure the TFA content were supplied in the revised Data S1. Standard deviation and t.test were also applied.

We have provided the dataset used to measure the TFA content. We found a TFA content of 236 mg (gC)-1 for TisoArg and 385 mg (gC)-1 for TisoS2M2 in 2019. This is similar to the measures reported by Bougaran et al. in 2012, which were 262 mg(gC)-1 and 409 mg (gC)-1 for TisoArg and TisoS2M2, respectively. The TFA content was calculated as the sum of PUFA, MUFA and SFA. This detail was added in Materials and Methods, and the dataset can be found in Data S1. We are also providing additional data of ƩΩ3 and ƩΩ6 content (revised Data S1 and Figure S1).

We agree that biological replicates, especially for model organisms, are commonly composed of n=3. However, the challenge and difficulty of working with the non-model microalga Tisochrysis lutea, constrained us to use two biological replicates. Nevertheless, the values of the two biological replicates are very similar, and the standard deviations are very low for OD and TFA content measures. 

We also applied a statistical test (t.test) which confirmed a significant difference between the TFA content of TisoS2M2 and TisoArg strains in 2019 (significant t. test with p< 0.05) (revised DataS1 and Fig. 1). As highlighted in the literature “there are no principal objections to using a t-test with Ns as small as 2.” from Winter, 2013.

Reference: [De Winter, Joost CF. "Using the Student's t-test with extremely small sample sizes." Practical Assessment, Research, and Evaluation 18.1 (2013): 10.]. https://doi.org/10.7275/e4r6-dj05

Consequently, we estimate that we have reliable data in our analysis.

Reviewer #1:Also, authors focus on TFA while they probably have more detailed results that could be displayed: the mat&meth describes FAMEs analyses and lipid separation. 

Our response: Our study focuses on TFA content because TisoS2M2 was previously described to have an increase in TFA content compared to TisoArg (Bougaran et al. 2012), with a similar growth rate. As requested by the reviewer, we provide additional results of the fatty acid analysis in the revised manuscript. We added MUFA, PUFA, and SFA measures used to calculate the TFA content (revised Data S1). We also provide data about ƩΩ3 and ƩΩ6 content (revised Data S1 and Figure S1b). 

Reviewer #1: By the way, there is ambiguity: are "TFA content" (line 91), "TFA ratio" (line 92) and/or “TFA productivity” (line 98) equivalent ? How are they computed from the raw lipid analyses?

Our response: The "TFA content" refers to the quantity in mgTFA(gC)-1, and "TFA ratio" refers to the difference in TFA quantity between TisoArg and TisoS2M2. We kept "TFA content" and "TFA ratio". We removed “TFA productivity” that we replaced with "TFA content". The TFA content was calculated from the sum of SFA, MUFA and PUFA. We added this detail in the Materials in Methods.

Reviewer #1: SNPs and indels in the native TisoArg (considered as wild-type, “WT” hereafter) as compared to reference genome are poorly commented and generally regarded as a “baseline”, while detected in a huge amount (line 114-115 : “18,451 […] SNPs were specific to TisoArg”). Are these SNPs “missed” in the mutant ? (neither SNP or indel predictors are perfect, it is a current challenge in the field, the authors may comment on that). 

Our response: As commented by the reviewer, we predicted SNPs and INDELs specific to the native TisoArg. Indeed, those TisoArg-specific SNPs and INDELs are predicted to be missing in the TisoS2M2 mutant. We agree with the reviewer that SNP or indel predictors are not perfect, and detecting such polymorphisms is challenging. 

We have commented on this point in the discussion as suggested:

“This analysis suggested that the improvement program induced changes at the single nucleotide level in TisoS2M2, leading to specific SNPs predictions in both strains. According to our analysis, SNPs did not seem to be the causal type of mutation associated with the lipid-enhanced phenotype of TisoS2M2. However, the fact that predictions of SNPs and their impacts on genes are challenging and that the knowledge about gene functions of T. lutea is limited, may have prevented the detection of a causal mutation.”

Reviewer #1:Or when/how they were acquired in the WT (in absence of UV treatment as in S2M2)? In my opinion, this is crucial to assume that indels and SNPs in S2M2 were caused by the mutagenesis. 

Our response: As commented by the reviewer, we don’t know when the SNP and INDELs were acquired in the Wt and the mutant. We also believe that most of them were induced by UV during the improvement program. We agree that it is essential to assume that the mutagenesis caused the specific SNPs and INDELs in TisoArg and TisoS2M2. We commented in the discussion: 

“While a part of genetic variants could have spontaneously appeared during the maintenance period, we assumed that the UV-based improvement program caused most of the specific SNPs and TE indels in TisoS2M2.”

We have also discussed in a paragraph about the potential effect of UV on the induction of TE activity.

Reviewer #1:Lines 115-119, Figure 1 and discussion should include information and interpretation on WT as well. 

Our response: We agree with the reviewer and we added the TisoArg-specific SNPs and INDELs detections. We illustrated results in the revised Figure 2 and 3 and provided the data in the revised Data S4. We have also added interpretations in the discussion.

Reviewer #1: Also, paragraph 5.5, can you explain why aligners bwa-mem and Mosaik are then used in conjunction with distinct indel/SNPs callers ? E.g. could you reinforce your detection of insertions by crossing results with Mobster on bwa aligned reads ?

Our response: We used two different aligners (bwa-mem and Mosaik) according to authors recommendation. 

For SNPs analysis, no aligner was required by DiscoSNP++ (https://github.com/GATB/DiscoSnp).

For deletion analysis, we used bwa aligner with both “Pindel” and “Breakdancer”, because this aligner is the first choice advised by their authors. https://gmt.genome.wustl.edu/packages/breakdancer/documentation.html and https://gmt.genome.wustl.edu/packages/pindel/user-manual.html

For the TE insertion analysis, we used Mosaik aligner with “Mobster” because this tools was developed to use this aligner. It is recommended to be used by the Mobster’s authors. (https://github.com/jyhehir/mobster). 

We did not use “Mobster” with bwa aligner because it could add false positives or true negatives as this tool was not developed to use bwa alignment.

Nevertheless, we also used long reads with “Sniffles” to reinforce our results by confirming numerous INDEL predictions (Data S4).

Reviewer #1: Information and interpretation derived from the Nanopore sequencing are relatively short. Methods in paragraph 5.3 could be more detailed (kits, basecalling mode, output quality and amounts,…). 

Our response: We agree with the reviewer and we added the requested information in the manuscript (kits, basecalling mode, minimum quality score and amounts of reads). 

We noticed an error in the measure of the sequencing coverage for long reads in the previous manuscript version. The corrected sequencing coverage is 4.1x for TisoArg and 7.9x for TisoS2M2.

Reviewer #1:The contradiction between several Nanopore reads is not clear. Is it different populations ? Is this locus heterozygous (diploidy/aneuploidy in S2M2?) ? 

Our response: As the reviewer commented, the strains TisoArg and TisoS2M2 are not clonal, but composed of populations, thought to be composed of diploid and haploid cells. Tisochrisis lutea is thought to have asexual and sexual life cycles as it has been described in other haptophytes. This would explain the fact that we retrieved a TE locus with heterozygosity (Figure S2 and revised Data S7). 

We commented on this point in the manuscript:

“Long read alignment made also possible to pinpoint a locus with heterozygous TE indel events (Figure S2 and Data S7). The life cycle of T. lutea is still unknown, but is potentially haploid-diploid as described for other haptophytes (67). The life cycle of T. lutea is uncontrolled at laboratory conditions and could lead to the presence of haploids and diploid cells in the algal population.”

Reviewer #1: Has the whole region been duplicated before transposon movement? Do these reads come from populations of cells that are distinct in this region ? 

Our response: As commented by the reviewer, we believe that the heterozygous TE locus (Figure S2 and Data S7) can be explained by the presence of cells that are distinct in this region.

Reviewer #1:Also, line 265 and after are slightly ambiguous. Do you mean that out 57 indels predicted from Illumina, 55 were not covered at all by Nanopore reads ? 

Our response: In the previous manuscript version, we used the tool “LorTE” to confirm INDELs predicted by tools using Illumina sequencing data. During this first analysis, LorTE only retrieved 2 out of 57 predictions, which led us to wrongly conclude that the 55 other indels were not covered by ONT reads or did not have the genetic variants. 

However, we now provide a new analysis using the tool “Sniffles” which greatly outperformed “LorTE” predictions. Sniffles retrieved and confirmed around half of the TE INDELs predicted by Illumina sequencing (Data S4 and Figure 3). We consequently removed “LorTE” analysis of the revised manuscript.

Reviewer #1: Or that the reads were inconclusive ? Or that Illumina and Nanopore were contradictory? This is probably crucial to your point. 

Our response: We agree that it is an important point. As mentioned above, we now provide a revised analysis of TE indel detection from long reads by using the tool “Sniffles”. In the revised manuscript, “Sniffles” confirmed around half the prediction obtained by tools using Illumina sequencing data. Moreover, the results are consistent between tools (Data S4). 

Reviewer #1: Have you tried to assemble de novo the WT and S2M2 genomes out of Nanopore reads (e.g using Flye, Canu, Smart de novo,…) and compare it three-way WT vs S2M2 vs reference contigs? (You may also polish these assemblies using Illumina reads).

Our response: In this study, we choose to use a strategy of identification of specific variants from a reference genome assembly. This strategy was applied previously and notably for the comparison of Wt and mutant microalgae (Schierenbeck et al. 2015 https://doi.org/10.1186/s12864-015-1232-y). We commented on this point in the discussion: 

“A previous study used a reference genome assembly and removed shared background variants between the wildtype strain and mutants to identify the causal mutations (27). We used a similar method to characterize the genetic changes (SNPs and TE indels) between TisoArg and TisoS2M2.”

Nevertheless, we understand the comment of the reviewer and we tried to build genome assemblies of TisoArg and TisoS2M2 from ONT reads using the assemblers Canu and WTDBG2. However, we obtained partial and fragmented genome assemblies (the expected genome size is 82 Mb) because of the little coverage of the ONT sequencing. We think these draft genome assemblies are unusable for TE indel analysis.

Please see the output summaries of Canu and WTDBG2 in the tables below.

Canu TisoArg TisoS2M2

Total Assembly length 7.5 Mb 60.6 Mb

Contig number 504 2,324

WTDBG2 Contigs >= 500 Largest contig Total length

TisoArg 1,935 256,648 43 Mb

TisoS2M2 1,189 559,646 77 Mb

Reviewer #1: Overall, this work could be re-shaped (and go deeper) to focus on the discovery of Shanks transposon family and mutation rates in T. iso (including description of WT SNPs and indels), and conclude on the phenotypic stability (with the proposed details). 

Our response: We agree with the reviewer and we performed additional analyses to go further. The manuscript was also deeply re-shaped following the suggestions of reviewers.

-We provide datasets of TFA content and OD and improved our analysis regarding the phenotypic stability of the lipid-enhanced phenotype of TisoS2M2. 

-We added analyses of specific SNPs and indels of TisoArg (WT) (revised Figure 2 and revised Data S4). We improved the analysis of TE INDELs by using ONT long reads with the tool “Sniffles” (revised Figure 3 and revised Data S4). 

-The new results obtained by “Sniffles” and ONT long reads were helpful to go deeper. We extracted sequences of specific TE insertions retrieved by Sniflles and confirmed that they were mostly caused by DNA transposons (revised Fig. 3). We also analyzed the type of TE involved (autonomous or non-autonomous TEs) (revised Fig. S3). We notably confirmed that most of the TE insertions caused by Shanks are related to non-autonomous elements.

The mutation rate of T. lutea is unknown which did not allow us to conduct this suggested analysis.

Reviewer #1: The candidate causal mutation could be an opening because it calls for more investigation in a later publication.

Our response: We agree with the reviewer and we commented in the discussion:

“Nevertheless, these two TE indels are predicted with moderate and low allele frequency (28 and 17%), and are probably not the main genetic(s) cause(s) of the lipid-enhanced phenotype of TisoS2M2. The selection of clonal strains having these TE indels would be an interesting proposition for further investigations. The causal mutation(s) of the TisoS2M2 phenotype remain to be explored.”

Reviewer #1: Data availability:

Will the Illumina and/or Nanopore sequences be made available and how?

Our response: The Illumina and Nanopore sequences are available online on SEANOE database https://doi.org/10.17882/83584

Minor remarks:

Reviewer #1: Some introductive elements are missing regarding S2M2: is this a “strain” (line 79) ? monoclonal ? Derived from a single cell at any step ? or is this a population (line 225) ? in the sense of being derived from the batch selection of distinct mutants ? or in the sense of genetic drift/emergence of new mutations in distinct cells during the 7 years cultivation which makes the culture closer to a population ?

Our response: TisoArg is a wildtype T. lutea strain, composed of a population of cells. TisoS2M2 is an improved strain also composed of a population of cells. TisoS2M2 was obtained by selecting the cells with the highest TFA content using a flow cytometer (Fig. 1) (Bougarant et al. 2012). These strains are maintained under laboratory culture conditions. It is possible that the strains have “naturally” evolved during the seven years of maintenance.

We added information in Materials and Method: “TisoArg and TisoS2M2 are not clonal but composed of populations of cells.”

Reviewer #1: Reading between the lines, one understands that no causal mutation(s) has been identified for the increased lipid production in S2M2 and that this works intends to find candidate genes, whether this is the case or not, this could be mentioned. 

Our response: We agree with the reviewer and we have re-shaped the manuscript to make this investigation clearer.

We have also mentioned in the revised discussion that no convincing causal mutation(s) has been identified:

“The causal mutation(s) of the TisoS2M2 phenotype remain to be explored.”

Reviewer #1: Regarding T. isochrysis in general, how large is the genome ? is it haploid/diploid ? if diploid, an idea of the level of heterozygosity ? Any hint whether S2M2 has conserved these features ?

Our response: The genome length is estimated to be 82 Mb (Berthelier et al. 2018). As mentioned above, T. lutea is thought to be capable of sexual and asexual reproduction as other haptophytes. To our knowledge, the T. lutea life cycle is still unknown.

Concerning the level of heterozygosity, we have no idea of the proportion. We believe both TisoArg and TisoS2M2 are composed of haploid and diploid cells. 

Reviewer #1: Could you estimate how many generations represent 7 years of cultivation ? Are there estimates of the mutation rate(s) in T. iso ? Can you comment regarding your observations ?

Our response: As commented above, the life cycle of T. lutea is still unknown. We cannot give an estimation of the number of generations. To the best of our knowledge, no estimation of mutation rate(s) in T. lutea has been established.

Reviewer #1:Line 98: “furthermore” => “In conclusion”

Our response: We thanks the reviewer for the comment. We corrected it.

Reviewer #1: Line 116: “hight” => “high”

Our response: We thanks the reviewer for the comment, but we have removed this sentence in the re-shaped manuscript.

Reviewer #1: Line 122: not sure of the syntax : “… enable to encode for …”

Our response: We thanks the reviewer for the comment, and we corrected it.

Reviewer #1: Table : line 2 and 3 have same locations, but distinct “gene ID”, can you comment ?

Our response: This is because two overlapping gene models were predicted in T. lutea genome at this locus (See Figure S5). We have commented it in the revised manuscript:

“The gene predicted to encode for the long-chain-enoyl-CoA hydratase has a length of 37,896 bp and seems to correspond to a false positive gene prediction. This gene prediction was discarded and not discussed further.”

Reviewer #1: Line 261: this challenges the view that DNA TEs are “cut-paste”, isn’t ? Is transposition independent from the encoded transposase then? 

Our response: As commented by the reviewer, the fact that DNA transposon mobility can occur during DNA replication or reparation, challenge the view of the “cut-and-paste” mechanism. Nevertheless, the transposition of the TEs is still processed by an encoded transposase. Indeed, autonomous TEs encode for their own transposase to move, but non-autonomous TEs dependent on the transposase encoded by autonomous TEs belonging to the same family to move. 

Reviewer #1: or is this observations rather due to sequencing/analyses limitations ? Since this work focuses on the transposition events, can you comment further ?

Our response: We agree that this observation could also be due to sequencing limitations and the difficulty to predict TE indels. The use of strains TisoS2M2 and TisoArg composed of populations of cells also increases the difficulty of this analysis. We propose to remove this sentence in the revised manuscript as we do not have much evidence about this point.

---------------

Reviewer #2: Transposable elements are important components of the eukaryotic genome. However, there are still a lot of research gaps in their genomic evolution, function and disease immunity. In this paper, the stability of lipid traits and genetic changes of TisoS2M2 were obtained by mutation-selection improvement program, and the effect of the improvement program on the TE kinetics of domesticated microalgae was determined for the first time. This study has certain innovation and reference, but some problems need to be explained.

Reviewer #2:1.Why use UV-induced induce TE mobility instead of other methods accepted by commercial and political rules? 

Our response: The obtention of TisoS2M2 was performed in 2010-2012, and at this time, the researchers who conducted this improvement program had little knowledge about the genetic background of the microalga. They used UV to generate random mutations and had no idea of the potential impact of this method on the TE activity.

We added a sentence about this point in the discussion:

“UV radiation is a robust and inexpensive method to generate random mutations and obtain a pool of mutants without previous knowledge of the genetics and metabolisms of the studied microalgae”

As commented by the reviewer, in the first submission of the manuscript, we added the following sentence in the discussion part which was misleading to the reader:

“Products of radiation mutagenesis are considered to be genetically modified organisms and are subject to commercial and political rules.”

GMO law is often changing, and we wrongly thought (at the time of the manuscript writing) that the improved strains obtained from UV were now submitted to commercial rules in Europe. Indeed, GMO definition was unclear for UV-based mutants a few years ago. 

After a more recent investigation, we found that in Europe, microalgae strains obtained through genetic manipulation are still subject to regulatory scrutiny. However, the Court of Justice of the European Union states that the GMO Directive (1) does not apply to organisms obtained by means of certain mutagenesis techniques, namely those which have conventionally been used (such as UV-mutagenesis) in a number of applications and have a long safety record (Judgment in Case C-528/16 - 2018). 

The Court nevertheless specifies that the Member States are free to subject such organisms, in compliance with EU law (in particular the rules on the free movement of goods), to the obligations laid down by the GMO Directive or to other obligations.

Consequently, this strain TisoS2M2 is accepted by commercial and political rules. Therefore, we removed the misleading sentence we wrote in the first manuscript.

(1) Directive 2001/18/EC of the European Parliament and of the Council of 12 March 2001 on the deliberate release into the environment of genetically modified organisms and repealing Council Directive 90/220/EEC

Reviewer #2: 2.The classification of transposable elements is also constantly changing, with the discovery of new element types being constantly revised and new classification scales being introduced. Methods and standards for identifying and categorizing transposable elements are also evolving, so whether there is a difference in the sequencing results from a few years ago.

Our response: As commented by the reviewer, new sub-group of TEs among the main Superfamilies are often introduced. However, the main Classes and Orders, and most of the Superfamilies, do not change. While new proposed classifications are from time to time published or introduced to add new classification scales, the two main followed classifications by the TE community are the one of “Wicker et al 2007” and the one of the database “Repbase” (Jurka et al.). 

The current annotation of T. lutea follows the Wicker et al 2007 classification system, which was used by the classification tool PASTEC (Hoede et al. 2014) we previously employed to classify the TE at the Superfamily level.

Indeed, we previously detected, classified, and annotated the TEs in the reference genome assembly of T. lutea using our own sophisticated pipeline (called “PiRATE”, see Berthelier et al. 2018). This pipeline is composed of the most efficient tools to conduct the detection and annotation of TEs (also presented in the review of Storer et al. 2022, DOI: 10.3390/genes13040709). Moreover, we built our own TE databank of microalgae in an effort to improve the TE classification step. 

We consider that the TE annotation of the reference genome of T. lutea is still very accurate.

Reviewer #2: 3.The article mentioned that some TE indels may affect genes involved in the neutral lipid pathway, but it did not verify which genes are changed in expression, whether it had a positive effect or a negative effect.

Our response: We agree with the reviewer that it would be an interesting analysis to perform. However, we do not have RNAseq data that we could use. Nevertheless, we agree that this investigation would be interesting to conduct in the future.

Reviewer #2: 4.In the "Results Section", the number 2.1 is followed by the number 2.3, and the number 2.2 is missing. Please correct it.

Our response: We thanks the reviewer for the comment. We corrected it.

Reviewer #2: 5.Tisochrysis lutea is not described in detail in Background, and what are the special features that led the author to study this type of algae

Our response: We agree with the reviewer and we added additional information about T. lutea in the Background.

Reviewer #2: 6.The reference format is not uniform, and the case is not standardized.

Our response: We thanks the reviewer for the comment. We have now uniformized the references and the case is standardized.

--------------

Reviewer #3: This study builds on previous work by the research group, describing changes in transposable elements in a strain that was improved to have improved lipid content through mutagenesis. The authors use the framework of domestication for the paper, which is appropriate and timely. The work is a first step to understand the mechanisms by which mutagenesis-selection regimes affect traits and/or trait stability. However, there are a few flaws that need attention. First (and glaringly), with the exception of their own work (reference 32), there is no discussion of previous work on transposable elements in algae. A decent amount of work has been done in Chlamydomonas and there is also relevant literature that can be pulled from the macroalgal world. Second, the paper seems to have been “stretched” with previous work. That is, methods and results of the previous work already published area included here. With a generous read, it is not necessary to do this and a shorter publication that explicitly states they are building off the former would still add to the body of literature. Third, the paper falls short of describing mechanisms in section 2.6. Finally, the paper needs restructuring (results are in the discussion, discussion is in the results). Specific comments are below.

Major comments

Reviewer #3: It seems all of the results in section 2.1 were published elsewhere. If this is the case, this content should be part of the materials and methods as the current study builds on the former. The results section should start with 2.2 (mislabeled 2.3 Prediction of polymorphism….) as that is the new work. I would also more explicitly state throughout the manuscript that this work builds on former publications.

Our response: Following to reviewer’s comment, we have rewritten section 2.1 to make it clearer that the results of the lipid from 2019 are new and were previously unpublished. In section 2.1, we have also reduced the details regarding the improvement program conducted in 2012 (Bougaran et al 2012). 

In 2.1: the main results are that we re-analyzed the TFA content of TisoArg and TisoS2M2 in 2019, and found that the domesticated strain TisoS2M2 still has a stable lipid-enhanced phenotype compared to the native TisoArg seven years after (2012 vs 2019). 

We have added more explicit sentences and references throughout the manuscript to make it clearer that the phenotypic analysis aimed to compare results from a previous publication (measure of 2012; Bougaran et al. 2012) and our new data (measures of 2019) (revised Fig. 1b).

Reviewer #3: Figure 1b is fine but not necessary, would consider whether this is worth including.

Our response: We would prefer to include Figure 1b to illustrate the phenotypic stability of TisoS2M2 over the years.

Reviewer #3: The second results section 2.2 (mislabeled 2.3) also starts with previously published work. This could also be moved to the methods section.

Our response: The results of section 2.2 had not been published elsewhere. We indicated that the Illumina and ONT sequencing (presented in sections 2.2. and 2.3) were both performed in 2018, because the lipid analysis was conducted in 2019. We believe the information of this timeline is important.

Reviewer #3: Lines 154-157: It should first be stated that 57 indels were identified, as later text refers to this number but it is easily glossed over here.

Our response: We agree with the reviewer. However, we have re-performed the analysis and the number of 57 indels have changed in the new revision. The discussion part has also been re-shaped as suggested by reviewers.

Reviewer #3: More references are needed for the sentence in line 213.

Our response: We agree with the reviewer and we added additional references.

“De novo TE insertions can affect genes to drive change in the host phenotype by affecting genes (35,36,41).”

Reviewer #3: Section 2.6 could hold the major crux of the paper, but there is very little content within it. The authors state the 31 insertions and the two deletions are located within or close to genes, and that nine were close to genes involved in lipid or catabolic metabolism. But only one deletion event is described. What about the other eight? There is some information on these in Table 1. Can these results be discussed further?

Our response: In the first manuscript version, we analyzed one by one the predicted functions of the other eight candidate genes (associated to “lipid” or “catabolic” GO terms) and searched in the literature for a potential role with the TFA regulation. However, we found no known role for homologous genes that we could discuss further. 

Here, we re-performed this search in the literature, and found a potential link for one additional candidate gene, encoding for a (Cyclin-dependent kinase A-1, Tiso_v2_10131). Indeed, a recent paper (from 2023) identified that a Cyclin-dependent kinase is involved in the regulation of the lipid content of Arabidopsis thaliana. We added this new detail in the revised manuscript.

However, we believe that the two main findings of our study are:

- Confirmation of the stability of the lipid-enhanced phenotype of TisoS2M2. Such analysis have never been conducted before and are lacking as highlighted by authors in the literature (such as the reviews of “Trovão et al 2022 in Marine Drugs”; or Jebali et al 2022 in Biotechnol Adv.). This is crucial in an industrial context to be sure that the improved phenotype of a domesticated strain is stable over time.

- The analysis of the dynamics of TEs in T. lutea and notably in the domesticated strains TisoS2M2. Our study reveals mobile TE families in T. lutea and suggests a potential impact of the improvement program on the TE activity. This is a significant finding that was never mentioned before and which is calling for new investigations in the future.

Reviewer #3: The authors bring up trait evolution. Although not related to lipid metabolism, I am curious about other traits that changed between the time of mutagenesis and present. Was there drift in other traits? Other trade-offs? There is room here for a broader analysis.

Our response: We agree with the reviewer that it is an interesting question. In addition to the lipid changes, we also analyzed the growth rate of TisoArg and TisoS2M2 and found no drift. However, we do not have additional data that we could use to investigate further trait evolution of TisoS2M2 between 2012 and 2019.

Reviewer #3: Reading through the results, I thought the paper was structured as an integrated results and discussion section. As they are separate, there is content that should be pulled from the results and placed in the discussion (e.g., lines 165-166; 168-169; lines 222-226). Similarly, the discussion section reads like a results section and is repetitive in many sections. For example, there is little to no new content in the first two pages of the discussion section. Both sections need to be considerably restructured.

Line 285, the discussion of the mutation-selection program, seems an appropriate place to start the discussion – and could be a quite interesting discussion point. Yet, content there is lacking.

Our response: We agree with the reviewer. We have deeply restructured the result and discussion parts following the reviewer’s recommendations.

Reviewer #3: The authors reference the use of UV mutagenesis in plants (one reference), but not algae. I would integrate the relevant algal literature. I might also expand the discussion on epigenetics a bit.

Our response: We agree with the reviewer, and we added a paragraph about UV mutagenesis in microalgae and we also extended the content about epigenetics.

Reviewer #3: The Conclusion includes prior work. Needs to be altered to consider only current work.

Our response: We do not present prior work in conclusion. We first introduce the lipid-enhanced strain TisoS2M2 (used in this study) for the reader’s understanding. We next present unpublished results on the phenotypic stability of the lipid-enhanced strain TisoS2M2. We also present the unpublished work on the genetic changes (TE dynamics) between the native TisoArg (Wt) and the domesticated TisoS2M2.

Reviewer #3: In the methods, Sections 5.1., 5.2, and 5.3 appear to be the previous work.

Our response: In section 5.1, we agree and reduced the description of the improvement program, performed in 2012 to obtain the domesticated strain TisoS2M2 (Bougaran et al 2012). 

However, we believe it is important to introduce the improvement program briefly. It is important to explain how the strains TisoArg and TisoS2M2 were obtained for the understanding of the manuscript. As mentioned above, sections 5.2 and 5.3 do not contain the results or conclusions of previous works.

We analyzed unpublished data from 2019 for the investigation of the phenotypic stability of the lipid-enhanced strain TisoS2M2 and we have also analyzed unpublished sequencing data from 2018 to describe the genetic changes between strains.

Reviewer #3: Lines 377 and 378: why were two programs used to map the assemblies? Could this have led to different artifacts?

Our response: We used two different aligners (bwa and Mosaik) according to the author’s recommendation. 

For deletion analysis, we used the bwa aligner with both “Pindel” and “Breakdancer”, because this aligner is the first choice advised by their authors. https://gmt.genome.wustl.edu/packages/breakdancer/documentation.html and https://gmt.genome.wustl.edu/packages/pindel/user-manual.html

For the TE insertion analysis, we used Mosaik aligner with “Mobster” because this tool was developed to use this aligner. It is recommended by the Mobster’s author (https://github.com/jyhehir/mobster). 

For SNPs analysis, no aligner was used by DiscoSNP++ (https://github.com/GATB/DiscoSnp).

The usage of different aligners did not cause any problems because the two alignments were used to identify different types of genetic variants. BWA alignment was used to identify deletion events and the MOSAIK alignment was used to detect insertion events.

Minor comments

Reviewer #3: The manuscript could use another read though to clean up grammar, sentence structure and flow. These stood out to me:

Lines 58-60: reference of “it” is misplaced, rephrase

Our response: We thanks the reviewer for the comment. We rephrased.

Reviewer #3: Line 64: the native strain

Our response: We thanks the reviewer for the comment. We changed it.

Reviewer #3: Line 67: which should be that, or comma should be added

Our response: We thanks the reviewer for the comment. We changed it.

Reviewer #3: Line 68: change they to transposable elements

Our response: We thanks the reviewer for the comment. We changed it.

Reviewer #3: Line 70: While should be although

Our response: We thanks the reviewer for the comment. We changed it.

Reviewer #3: Line 82: compared to

Our response: We thanks the reviewer for the comment. We changed it.

Reviewer #3: Line 116: typo in hight

Our response: We thanks the reviewer for the comment. We changed it.

Reviewer #3: Line 167: compared

Our response: This sentence was removed in the re-shaped manuscript.

Reviewer #3: Line 168: this result suggests

Our response: This sentence was removed in the re-shaped manuscript.

Reviewer #3: Line 179: delete around, here and in other places; or replace with approximately if the bp number is approximate (it seems the number is precise, so may also change bp number)

Our response: We thanks the reviewer for the comment. We changed it.

Reviewer #3: Line 199: add comma after study

Our response: We thanks the reviewer for the comment. We changed it.

Line 200: add comma after TIR/hATv

Our response: We thanks the reviewer for the comment. We changed it.

---

## [Decision Letter · Decision Letter 1]

5 Apr 2023

Phenotype stability and dynamics of transposable elements in a strain of the microalga Tisochrysis lutea with improved lipid traits

PONE-D-22-30286R1

Dear Dr. Berthelier,

We’re pleased to inform you that your manuscript has been judged scientifically suitable for publication and will be formally accepted for publication once it meets all outstanding technical requirements.

Kind regards,

Balamurugan Srinivasan

Academic Editor

PLOS ONE

Additional Editor Comments (optional):

Reviewers' comments:

Reviewer's Responses to Questions

**Comments to the Author**

1. If the authors have adequately addressed your comments raised in a previous round of review and you feel that this manuscript is now acceptable for publication, you may indicate that here to bypass the “Comments to the Author” section, enter your conflict of interest statement in the “Confidential to Editor” section, and submit your "Accept" recommendation.

Reviewer #2: All comments have been addressed

Reviewer #3: All comments have been addressed

2. Is the manuscript technically sound, and do the data support the conclusions?

Reviewer #2: Yes

Reviewer #3: Yes

3. Has the statistical analysis been performed appropriately and rigorously? 

Reviewer #2: Yes

Reviewer #3: Yes

4. Have the authors made all data underlying the findings in their manuscript fully available?

Reviewer #2: Yes

Reviewer #3: Yes

5. Is the manuscript presented in an intelligible fashion and written in standard English?

Reviewer #2: Yes

Reviewer #3: Yes

6. Review Comments to the Author

Reviewer #2: Thank you very much for the author's careful reply to my questions. I think this paper can be accepted and considered for publication

Reviewer #3: The authors have responded to my comments and the comments of the other reviewers. Although not every recommendation was followed, the paper is appropriate for publication.

7. PLOS authors have the option to publish the peer review history of their article (what does this mean?). If published, this will include your full peer review and any attached files.

Reviewer #2: No

Reviewer #3: No

---

## [Editor Report · Acceptance letter]

19 Apr 2023

PONE-D-22-30286R1 

Phenotype stability and dynamics of transposable elements in a strain of the microalga *Tisochrysis lutea* with improved lipid traits 

Dear Dr. Berthelier:

I'm pleased to inform you that your manuscript has been deemed suitable for publication in PLOS ONE. Congratulations! Your manuscript is now with our production department. 

Kind regards, 

on behalf of

Dr. Balamurugan Srinivasan 

Academic Editor

PLOS ONE